# The Gilded Clot: Review of Metal-Modulated Platelet Activation, Coagulation, and Fibrinolysis

**DOI:** 10.3390/ijms24043302

**Published:** 2023-02-07

**Authors:** Vance G. Nielsen, Tanner Goff, Brent D. Hunsaker, Coulter D. Neves

**Affiliations:** Department of Anesthesiology, The University of Arizona College of Medicine, Tucson, AZ 85724, USA

**Keywords:** platelets aggregation, coagulation, fibrinolysis, metals, metal poisoning, thrombelastography, prothrombin time, activated partial prothrombin time, fibrinogen

## Abstract

The processes of blood coagulation and fibrinolysis that in part maintain the physical integrity of the circulatory system and fluidity of its contents are complex as they are critical for life. While the roles played by cellular components and circulating proteins in coagulation and fibrinolysis are widely acknowledged, the impact of metals on these processes is at best underappreciated. In this narrative review we identify twenty-five metals that can modulate the activity of platelets, plasmatic coagulation, and fibrinolysis as determined by in vitro and in vivo investigations involving several species besides human beings. When possible, the molecular interactions of the various metals with key cells and proteins of the hemostatic system were identified and displayed in detail. It is our intention that this work serve not as an ending point, but rather as a fair evaluation of what mechanisms concerning metal interactions with the hemostatic system have been elucidated, and as a beacon to guide future investigation.

## 1. Introduction

The processes of blood coagulation and fibrinolysis that in part maintain the physical integrity of the circulatory system and fluidity of its contents are complex as they are critical for life. While the roles played by cellular components and circulating proteins in coagulation and fibrinolysis are widely acknowledged, the impact of metals on these processes is at best underappreciated. Specifically, there are twenty-five metals that can modulate the activity of platelets, plasmatic coagulation, and fibrinolysis [1,2,3,4,5,6,7,8,9,10,11,12,13,14,15,16,17,18,19,20,21,22,23,24,25,26,27,28,29,30,31,32,33,34,35,36,37,38,39,40,41,42,43,44,45,46,47,48,49,50,51,52,53,54,55,56,57,58,59,60,61,62,63,64,65,66,67,68,69,70]. This large number of metals may be somewhat surprising to most, as typically most clinicians and scientists in this space would easily acknowledge the role of calcium as a critical cofactor in coagulation and fibrinolysis [9,10,11,12,13,14] but be pressed to quickly identify the role of other metals in these processes. However, as humans are exposed to metals in a variety of ways, such as endogenous metabolism (e.g., iron released during heme catabolism), environmental/industrial exposures (e.g., cadmium), clinical therapies (e.g., lithium to treat depression) and dietary supplements (e.g., vitamins with iron, zinc, etc.), laboratory and clinical manifestations of metal-associated changes in coagulation have been either recognized or serendipitously identified. Therefore, given the complexity of the interactions of metals with platelet activity, plasmatic coagulation and fibrinolysis, the goal of this narrative review is to present what is and is not known about these interactions to assist basic scientists and clinicians in future investigations and clinical care.

## 2. Definitions and Search Strategy

While perhaps considered a facile matter, the criteria allowing the consideration of specific metals to include in this review needed to be based on the likelihood that the unique cation formed by the metal and associated ions would modulate other cells/molecules in a metal-specific fashion. Examples of such metals are displayed in Figure 1, panel A, consisting of compounds containing iron, calcium, platinum, and gallium. The release of one or more chloride ions would be possible for calcium, iron or platinum, allowing for metal-target protein interactions. Similarly, the release of one or more nitrate ions from gallium would be possible to facilitate metal-coagulation protein binding. However, in the cases of the metalloid arsenic (As) or the metal tungsten (W), the cations of interest are bound to oxygen without exposure of the target molecule secondary to direct interaction with arsenic or tungsten as displayed in Figure 1, panel B. Thus, while both arsenic- and tungsten-containing compounds have been found to diminish platelet activity [71,72], it is the unique molecular moiety, not the metal, that is most likely responsible for antiplatelet effects. Similarly, while bismuth (Bi) is part of the hemostatic agent bismuth subgallate that activates factor XII (FXII) [73,74], Bi is bound to a modified gallic acid, a sizable organic molecule. It is unlikely that bismuth would be released from its organic conjoiner, and the structure of ellagic acid (another FXII activator) is like bismuth subgallate [73]. Thus, as with arsenic and tungsten, bismuth is most likely part of a unique hemostasis-modulating molecule but not a primary molecular cause of pro- or antihemostatic effects. Other situations involving metals and clotting function include the effects of ionizing radiation from metals such as uranium on coagulation [75], or the FXII-activating activity of metals such as titanium that are not in solution but are part of a biomaterial surface [76]. In summary, we identified metal salts most likely to influence platelet activity, coagulation, and fibrinolysis secondary to metal-protein interactions.

As our search strategy, we combined the metal of interest with “platelet activity or aggregation,” “coagulation” or “fibrinolysis” to obtain the literature to be presented [1,2,3,4,5,6,7,8,9,10,11,12,13,14,15,16,17,18,19,20,21,22,23,24,25,26,27,28,29,30,31,32,33,34,35,36,37,38,39,40,41,42,43,44,45,46,47,48,49,50,51,52,53,54,55,56,57,58,59,60,61,62,63,64,65,66,67,68,69,70]. For simplicity, the metals and their effects on platelets, coagulation and fibrinolysis are subsequently presented in alphabetical order based on the periodic table symbol of each metal. Further, the text of each metal will present what was captured with the search strategy, so if there is no information provided concerning the specific metal and platelet activity, coagulation, or fibrinolysis, we have not noted what was not found for each metal. Lastly, immediately following the subsequently presented text concerning the metals, Figure 2 depicts the metals presented as activators or inhibitors of platelet activity, Figure 3 indicates where the metals affect the various parts of the plasmatic coagulation pathways, and Figure 4 indicates where the metals affect the key enzymes of the fibrinolytic system.

As a final note before the metals are presented, a significant effort was made to determine what, if any, blood, or plasma concentrations of the metals are present in the settings of normal living or poisoning. In the case of the in vitro studies, the concentrations utilized by investigators in decades past may seem very large and may not correspond to reasonable in vivo or cellular concentrations. Nevertheless, the authors hold that presentation of what is known is of greater benefit than post hoc judging experimentation from the past.

## 3. Specific Metals and their Effects on Platelet Activation, Coagulation, and Fibrinolysis

### 3.1. Al—Aluminum (Atomic Number 13, Atomic Weight 26.98)

Al toxicity can be encountered a variety of ways secondary to exposure to contaminated air, water, medicines, antiperspirants, and industrial processes [77]. The multiorgan symptoms associated with aluminum toxicity have been attributed to free radical formation by its various salts [77].

With regard to effects on platelet aggregation, investigation was undertaken to determine if aluminum would effect platelet activity via generation of radical formation [1]. Isolated, human platelets were exposed to AlCl_3_ in a range of 0–100 µM, with aggregation observed beginning at 50 µM [1]. Dose-dependent, aluminum-mediated lipid peroxidation was detected by chemiluminescence, and both platelet aggregation and chemiluminescence following aluminum exposure was abrogated with 100 µM of either of the antioxidants nor-dihydrogaiaretic acid and n-propyl gallate [1]. Pretreatment of the platelets with acetyl salicylic acid decreased aluminum-mediated chemiluminescence and aggregation, supporting the hypothesis that aluminum activates platelets via the lipoxygenase pathway [1].

In summary, aluminum is a platelet activator in vitro via free radical generation.

### 3.2. Au—Gold (Atomic Number 79, 196.97 Atomic Weight)

Au has caused complex toxicity either secondary to exposure as jewelry or as medicinal compounds used for inflammatory disorders such as rheumatoid arthritis [78].

With regard to effects on platelets, AuCl_3_ was found to inhibit adenine diphosphate (ADP) and collagen-mediated aggregation in rat platelets [2].

Further, coagulation in rat plasma was inhibited in a dose-dependent manner, with AuCl_3_ (50–500 µg/mL; 164 µM−1.64 mM) demonstrating a direct inhibition of thrombin and modification of fibrinogen to reduce its clottability [2]. Gold sodium thiomalate (C_4_H_3_AuO_4_S_2_) and other gold compounds were found to decrease thrombus formation in vivo in a rabbit model secondary to platelet-independent mechanisms, most likely secondary to inhibition of thrombin [3]. The authors postulated that C_4_H_3_AuO_4_S_2_ disrupts disulfide bridges found in thrombin that are critical to enzymatic function [3].

With regard to fibrinolysis, C_4_H_3_AuO_4_S_2_ was not found to have any profibrinolytic effects in the rabbit [3] or in human whole blood in vitro [79]. However, when medically administered in vivo, blood fibrinolysis was increased in patients administered C_4_H_3_AuO_4_S_2_, and posited to be secondary to a decrease in alpha-2-globulins [4].

Taken as a whole, gold appears to inhibit platelets in either a species-specific or compound specific manner [2,3], but gold salts have been found to be antithrombotic regardless of compound or species [2,3]. Lastly, while C_4_H_3_AuO_4_S_2_ may not be a direct profibrinolytic agent [79], it appears that chronic treatment with this compound enhances fibrinolysis in vivo during chronic therapy [4].

### 3.3. Ba—Barium (Atomic Number 56, Atomic Weight 137.33)

Barium toxicity can occur following exposure in industrial settings or after exposure to insecticides, rodenticides, fireworks or depilatories, causing severe gastrointestinal symptoms and potentially paralysis [80].

With regard to effects on rat platelets, concentrations of 0.33–2.0 mM BaCl_2_ caused dose-dependent increases in aggregation, but 5.0–10.0 mM concentrations resulted in decreased aggregation [5].

As for coagulation, in decalcified rabbit plasma exposed to BaCl_2_, the thrombin-mediated clotting time is increased following addition of CaCl_2_ [6]. Further, when the decalcified plasma was clotted with Russell’s viper venom instead of tissue factor, the same pattern of decreased clotting time at small concentrations and increased clotting time with greater concentrations was observed with BaCl_2_ [6]. However, when plasma was exposed to a greater degree of decalcification, BaCl_2_ no longer affected clotting time [6]. Thus, it was posited that barium was releasing plasma protein-bound Ca, resulting in the observed decreased clotting times [6]. Also of note, BaCl_2_ did not affect bovine fibrin monomer polymerization between 1–1000 µM [7].

In summary, barium enhances platelet aggregation through undefined mechanisms, while likely only affecting plasmatic coagulation via release of Ca bound to plasma proteins.

### 3.4. Be—Beryllium (Atomic Number 4, Atomic Weight 9.01)

Beryllium toxicity can be found in manufacturing settings, such as electronic, atomic and defense industries, with severe pulmonary disease its sine quo non [81]. As for effects on platelet aggregation, BeCl_2_ (2.5–10 µg/mL; 31.3–125.2 µM) was found to enhance activation by arachidonic acid (AA) but not by activation by adenosine diphosphate (ADP) or collogen [8]. The mechanism underlying this BeCl_2_-mediated enhancement of activation was found to be secondary to decreasing the threshold to arachidonic acid and enhancing the production of thromboxane B_2_ (TXB_2_) [8]. In summary, beryllium enhances AA-mediated platelet aggregation.

### 3.5. Ca—Calcium (Atomic Number 20, Atomic Weight 40.08)

Rather than being considered an element toxic to hemostasis, calcium is critical to the normal functioning of platelets, plasmatic coagulation and fibrinolysis—it is also an essential element [5,7,9,10,11,12,13]. This is why whole blood and plasma samples undergo ethylenediaminetetraacetic acid (EDTA)- or sodium citrate-mediated chelation of calcium to inhibit critical Ca-dependent enzymatic or cellular-mediated coagulation prior to laboratory analyses.

With regard to platelets, calcium is required for microtubular formation, exocytosis of granular contents, and assembly of the αIIbβ3 receptor on the exterior membrane to bind to von Willebrand factor and fibrin polymers to facilitate the major portion of contractile resistance of formed clots [10,12]. The reader is referred to [10] for greater detail concerning calcium and activation of platelet function.

With regard to plasmatic coagulation, calcium is required for activation of coagulation factor XI, factor X, factor VIII, factor VII, factor V, and factor II (FII) (prothrombin), forming factor IIa (thrombin) [12]. Thrombin, using calcium as a cofactor, polymerizes fibrinogen to fibrin, activates platelets, and activates factor XIII to crosslink fibrin polymers [12].

In the matter of fibrinolysis, calcium is globally required to allow fibrinolysis to proceed [13], and it has been found to play a critical role in key fibrinolytic enzyme activities [82,83,84]. Specifically, calcium enhances plasmin-mediated lysis in human plasma, contributing to profibrinolytic effects [13]. With regard to antifibrinolytic effects, calcium is critical to stabilizing the key antifibrinolytic enzyme, plasminogen activator inhibitor type 1 (PAI-1) within platelet α-granules, allowing inhibition of tissue type plasminogen activator (tPA) following platelet activation [82]. Further, the potent antifibrinolytic enzyme, thrombin-activatable fibrinolysis inhibitor (TAFI), is activated by either thrombin or plasmin in a Ca-dependent fashion [83]. Lastly, the binding of the most potent plasmin inhibitor, α2-plasmin inhibitor, to fibrin polymers to oppose the effects of plasmin during fibrinolysis is also calcium-dependent [84]. In conclusion, calcium plays pivotal roles in platelet activation, coagulation, and fibrinolysis.

### 3.6. Cd—Cadmium (Atomic Number 48, Atomic Weight 112.41)

Cadmium toxicity has increased secondary to industrial development and has entered the environment via water, atmosphere, food and soil sources [85]. The metal accumulates in liver, bone, kidney and other organs, causing irreversible damage over time [85].

In the case of platelet aggregation assessed with platelet-rich plasma (PRP), cadmium (5 mM) can induce aggregation by displacing and substituting for Ca as the stimulus [14]; further, the platelets activated by cadmium appear “less rigid” than those activated by Ca as determined with scanning electron microscopy (SEM) [14]. In a more recent study involving exposure of whole blood and PRP to CdCl_2_ (26 µM), SEM analyses demonstrated altered platelet morphology consistent with increased activation of platelets [15].

With regard to coagulation, thrombelastographic analyses under the same conditions demonstrate a mild prolongation to onset of clot formation, decrease in velocity of clot growth and decrease in clot strength after exposure of whole blood to cadmium compared to control conditions; however, these changes did not reach statistical significance [15]. In a system of bovine fibrin monomers containing Ca chelators, CdCl_2_ (100–1000 µM) caused significant polymerization [7], likely secondary to binding to the carboxylic acid groups of the fibrin monomers.

As for effects on fibrinolysis, cadmium modulated the release of tPA and PAI-1 from cultured human smooth muscle cells and fibroblasts [16]. In the case of smooth muscle cells, cadmium (0.5–5 µM) significantly decreased both tPA and PAI-1 release [16]. In contrast, cadmium (1–5 µM) caused a significant increase in tPA release but no change in PAI-1 release from human fibroblasts [16]. Further, cadmium (up to 20 µM) induces increased PAI-1 expression without affecting tPA expression in a non-cytotoxic manner [17]; further, this cadmium-associated induction of PAI-1 is in part mediated by transcriptional factors, mothers against decapentaplegic homolog 2 and 3 (Smad2 and Smad3) [17].

In conclusion, cadmium appears to cause platelet aggregation at fairly large concentrations but tends to damage platelet structure at far smaller concentrations [14,15]. As for coagulation, while fibrin polymerization may be enhanced by cadmium [7], whole blood coagulation is mildly inhibited [15]. Lastly, on the cellular level, cadmium appears to have antifibrinolytic effects at the endothelial cell level in terms of increased PAI-1 expression [17], but cadmium also appears to exert a profibrinolytic effect via increased tPA release from fibroblasts [16].

### 3.7. Co—Cobalt (Atomic Number 27, Atomic Weight 58.93)

Cobalt is encountered in drinking water, vegetables and fish, with industrial exposures also noted—it is an essential element [86]. Cobalt exposure that is large and acute or chronic can cause adverse effects that may include the thyroid gland (inhibition of tyrosine iodinase, inducing goiter and potentially myxedema), the lungs (“cobalt asthma”), the skin (contact dermatitis) and the immune system, and may include a possible carcinogenic potential [86].

With regard to effects on human platelet aggregation, CoCl_2_ (3.85 mM) reduced ADP-induced aggregation by 70% in PRP exposed to cobalt for 90 min [18]. However, when PRP obtained from an afibrinogenemic patient was exposed to cobalt for 90 min and had Co-naïve fibrinogen added, aggregation was restored [18]. This novel finding indicated that cobalt was modifying fibrinogen and not the platelets when inhibiting platelet aggregation [18]. With regard to coagulation, CoCl_2_ (100–1000 µM) caused significant polymerization of bovine fibrin monomers, likely secondary to binding to the carboxylic acid groups of fibrin [7]. Taken as a whole, cobalt prevents fibrin from interacting with platelets, decreasing platelet aggregation in an indirect manner while also perhaps enhancing plasmatic coagulation by enhancing fibrin monomer polymerization.

### 3.8. Cr—Chromium (Atomic Number 24, Atomic Weight 51.99)

Chromium can be found in drinking water and in industrial settings involving numerous processes, including leather tanning, chrome pigment production, stainless steel manufacturing and chrome plating [87]. Chromium exerts its toxicity primarily as a carcinogen [87].

In the matter of platelet aggregation, chromium (985 µM) altered SEM morphology of platelets in PRP, consistent with increased activation [15].

With regard to coagulation, however, whole blood thrombelastography demonstrated that chromium exposure tended to prolong the time to onset of clot initiation, decreased the velocity of formation and decreased clot strength, but not statistically significantly [15]. Further, SEM of the fibrin network of clots formed after chromium exposure demonstrated less taut or bent fibers, characteristic of weaker clots [15].

Lastly, chromium exposure has been associated with pulmonary cancer and fibrosis, states that feature decreased urokinase-type plasminogen activator (uPA) and fibrinolysis as recently reviewed [19]. It was determined that exposure of alveolar type II cells in culture to 1 and 5 µM chromium resulted in decreased specific activity and amount of uPA protein. Chromium reduced uPA protein levels by inhibition of protein synthesis without any effect on mRNA levels [19].

In conclusion, chromium appears to potentially enhance platelet activation by SEM but overall mildly suppresses coagulation. In contrast, chromium may exert antifibrinolytic effects in the setting of inflammation and cancer by decreasing uPA concentrations.

### 3.9. Cs—Cesium (Atomic Number 55, Atomic Weight 132.90)

Cesium toxicity has been noted when it has been administered as an alternative medicinal agent for cancer or as a preventative to avoid injury from its isotope ^137^Cs in the setting of radioactive accidents such as in Chernobyl [88]. Symptoms include gastrointestinal distress, generalized numbness, hypotension, loss of consciousness, and prolonged QTc interval [88].

With regard to effects on human platelets, CsCl (50 mM) significantly inhibited ADP-mediated aggregation of heparin-anticoagulated PRP [20].

In the case of coagulation, in vitro, CsCl had no significant effect on factor IX (FIX) activation compared to a variety of monovalent salts; therefore, cesium does not affect FIX activity [21].

In conclusion, the only notable effect of cesium is to inhibit ADP-mediated activation of platelets.

### 3.10. Cu—Copper (Atomic Number 29, Atomic Weight 63.54)

Chronic exposure to inorganic copper has been hypothesized to contribute to Alzheimer’s disease [89] despite being an essential element, and acute, large exposures that are accidental or suicidal in nature have caused multiorgan failure with gastrointestinal hemorrhage being common [90].

In the matter of platelet activation, CuSO_4_ (1.9–6.3 mM) inhibited aggregation of rat platelets in PRP induced by ADP or collagen [2]. In contrast, in concentrations between 0.001–100 mM, Cu^2+^ (unspecified if Cl or SO_4_ salt) increased human platelet attachment to albumin, plastic collagen or fibrinogen (the data were not shown in the source reference) [22].

As for coagulation, CuSO_4_ (1.9–6.3 mM) was noted to exert an anticoagulant effect, characterized by at first enhancement and then inhibition of purified rat thrombin activity (perhaps based on increasing concentration of copper) and a minimal impairment in the clottability of rat fibrinogen [2]. In sharp contrast, it was demonstrated that 0–1000 µM CuCl_2_ resulted in a concentration-dependent loss of velocity of clot growth and strength determined by thrombelastography [23]. Further, by using a variety of clotting factor deficient plasmas and other biochemical techniques, it was determined that the mechanism underlying the anticoagulant effect of CuCl_2_ was likely disruption of disulfide bridges in fibrinogen [23]. However, purified bovine fibrin monomers in Tris-HCl buffer exposed to the chelating agent ethylenediaminetetraacetic acid (EDTA) were noted to polymerize in the presence of 10–1000 µM Cu^2+^ [7].

As for effects on fibrinolysis, CuSO_4_ enhanced streptokinase-mediated lysis of gelatin at a concentration of 0.5 parts per million (ppm) but diminished fibrinolysis at 2.0 ppm [24]. Fibrinolysis with streptokinase was also observed when human fibrinogen clots were exposed to CuSO_4_ at 0.0125–20 ppm with similar results of enhancement followed by complete inhibition of fibrinolysis [24]. CuSO_4_ at 215 µM was found to cause PAI-1 to assume a latent and not active configuration, a molecular event that would promote fibrinolysis [25].

In summary, while large concentrations of copper may enhance human platelet aggregation, smaller concentrations of copper would already have compromised coagulation by damaging fibrinogen and potentially enhancing fibrinolysis by PAI-1 inhibition.

### 3.11. Fe—Iron (Atomic Number 26, Atomic Weight 55.84)

Iron toxicity chronically occurs in a variety of hereditary and acquired diseases, such as hereditary haemochromatosis, sickle cell anemia, α- and β-thalassemia, and end-stage kidney failure treated with hemodialysis—but it is an essential element [91]. In these conditions, the enzyme heme oxygenase-1 catabolizes heme to biliverdin, carbon monoxide, and iron [91]. In chronic disease, multiorgan failure is observed from excessive deposition of iron in intracellular stores [91]. Acute ingestion of iron in the form of dietary supplements is unfortunately common in the pediatric population, presenting with severe gastrointestinal symptoms that can progress to shock and death [91].

The anticoagulant or procoagulant effects of iron containing salts potentially appear species, concentration, and iron valance dependent. As for effects on platelet aggregation, FeSO_4_ inhibited rat platelet aggregation induced by ADP at 13.2–32.9 mM, whereas aggregation caused by collogen was inhibited by 26.3–32.9 mM FeSO_4_ [2].

With regard to plasmatic coagulation, exposure of either rat thrombin or fibrinogen to 26.3–32.9 mM FeSO_4_ resulted in a loss of activity and clottability, respectively [2]. An early work demonstrated that with 1.3–8.1 mM FeSO_4_ to mM (posited to be similar to that seen during acute poisoning), coagulation was compromised in human plasma in vitro, primarily secondary to inhibition of thrombin activity—which was reversible with chelation [92]. Importantly, there was no direct effect of FeSO_4_ on thrombin-mediated fibrinogen polymerization as determined by thrombin time [92]. In sharp contrast, at concentrations varying from 0–10 µM up to 15 mM, depending on the methodology utilized, Fe^3+^, but not Fe^2+^, exposure results in enhanced coagulation kinetics and ultrastructural change in fibrin polymerization [26,27,28,29,30,31,32,33,34,35]. SEM-based investigations demonstrated progressive matting and connections of fibrin polymers after exposure to FeCl_3_ compared to unexposed fibrinogen or plasma [26,27,28,29,30,31,32,33]. Coagulation kinetics observed in human plasma exposed to FeCl_3_ can be characterized with a decrease in time to clot initiation and increase in the velocity of clot growth compared to plasma not exposed to FeCl_3_ [32,33,34,35]. Of interest, Fe^3+^-mediated changes in clot ultrastructure were abrogated by pretreatment of fibrinogen and plasma with antioxidants effective against hydroxyl radical or chelators that bind Fe^3+^ [29,30]. The presumed mechanism was the attenuation or prevention of hydroxyl radical formation and modification of fibrinogen [29,30]. However, complementary studies demonstrated that Fe^3+^-mediated ultrastructural and coagulation kinetic changes could be partially reversed by addition of deferoxamine to human plasma after Fe^3+^ exposure, revealing a reversible mechanism of hypercoagulability in contrast to irreversible hydroxyl radical modification of fibrinogen [33,34]. An additional study localized the binding site of Fe^3+^ on fibrinogen to be its alpha chain [35]. The clinical presence of iron-mediated hypercoagulability has been demonstrated in inflammatory conditions involving HO-1 upregulation as assessed with the aforementioned thrombelastograph-based chelation assay [33,34], which include calcific valvulopathy [93], Alzheimer’s disease [94], ventricular assist device therapy [95], sickle cell disease [96], and chronic migraine headache [97].

As for fibrinolysis, 10 µM FeCl_3_ increases clot lifespan and clot lysis time in a viscoelastic model using human plasma with tPA added [32]. Further, the coapplication of the metal chelator EDTA and tPA enhance fibrinolysis of murine thrombi in vitro and in vivo [36]. The molecular mechanisms responsible for resistance to tPA-mediated fibrinolysis in the presence of excess Fe^3+^ may include decreased access to fibrin polymer lysine residues in a denser system of fibers in vitro [32]; however, it is unclear how the coapplication of EDTA and tPA result in superior thrombolysis in vivo compared to tPA alone with physiological Fe [36].

In conclusion, Fe^2+^ appears to exert antiplatelet and antithrombotic effects acutely in the mM range, whereas Fe^3+^ causes procoagulant and antifibrinolytic effects both acutely and chronically in the µM range.

### 3.12. Ga—Gallium (Atomic Number 31, Atomic Weight 69.72)

Gallium has been used as a chemotherapeutic agent against several malignancies such as lymphomas and bladder cancer, with gallium nitrate (Ga(NO_3_)_3_) most commonly administered [98]. Toxicity from prolonged infusion of Ga(NO_3_)_3_ involves the intestine, manifesting as diarrhea [98].

Of interest, Ga(NO_3_)_3_ was initially found to exert local hemostatic effects and was subsequently investigated to determine the mechanisms responsible for these effects [37]. With regard to plasmatic coagulation, 7.5–15 mM Ga(NO_3_)_3_ diminishes whole blood clot strength by over 50% as determined by thrombelastography; further 39–50 mM Ga(NO_3_)_3_ removed clottable fibrinogen in human plasma [37]. No other significant effects on coagulation kinetics (e.g., onset time of coagulation) were observed [37]. Visual inspection of the plasma samples demonstrated precipitated material that the authors postulated was flocculation of fibrinogen by Ga(NO_3_)_3_ [37]. Thus, the mechanisms responsible for the purported hemostatic effects of gallium in vivo remain to be elucidated.

### 3.13. Hg—Mercury (Atomic Number 80, Atomic Weight 200.59)

Mercury exposure from environmental sources is ubiquitous [99]. Mercury-containing compounds can be inhaled, ingested from sources as diverse as dental amalgam, industrial release, or consumption of fish [99]. Multiple organs can be damaged by mercury-containing compounds, including the lungs, kidneys, and the nervous system [99].

With regard to effects on platelets, 29.9 nM mercury activated human platelets as evidenced by formation of pseudopodia and membrane spreading assessed with SEM [40]. The concentration of mercury chosen was based on the maximum allowed by the World Health Organization [40]. The source and salt of the mercury compound used was not mentioned [40].

As for coagulation, exposure of citrated, human whole blood to 29.9 nM mercury prior to coagulation resulted in a fibrin network of thick, thin and less taut fibers compared to unexposed blood samples [40]. In a rat model of acute HgCl_2_ poisoning (17.9 mg/kg, intragastric administration), over a 7-day period the animals had a marked increase in coagulation [38]. This was manifested as a significant decrease in the time to onset of coagulation, increase in the velocity of clot growth and increase in clot strength [38]. Plasma fibrinogen concentrations tripled [38]. In contrast to in vivo investigation, exposure of citrated human plasma to 50 µM HgCl_2_ (a near-maximum concentration observed in lethal poisoning) for 5 min prior to onset of thrombelastographic monitoring resulted in no significant differences in coagulation kinetics compared to HgCl_2_ naïve samples [39].

In the case of fibrinolysis, the aforementioned rat model demonstrated significant antifibrinolytic effects after acute HgCl_2_ poisoning [38], manifested as a profound increase in euglobulin lysis time accompanied by loss of plasminogen activator and plasmin activity in the plasma [38]. In human cell culture, 1 µM Hg decreased tPA and PAI-1 release from vascular smooth muscle cells and fibroblasts [16]. Lastly, in a very early work, mercury-containing compounds, including p-chloromecuribenzoate (C_7_H_5_ClHgO_2_) and meralluride (C_16_H_23_HgN_6_NaO_8_), were demonstrated to paradoxically restore plasmin activity inhibited by soybean trypsin inhibitor at µM-mM concentrations [41].

Considered as a whole, the aforementioned investigations support the concepts that mercury at small concentrations may activate platelets but does very little to either the coagulation or fibrinolytic pathways. In contrast, the in vitro and in vivo changes in coagulation observed in cell culture and mercury-poisoned rats, respectively, likely reflect indirect effects of Hg on coagulation and fibrinolysis secondary to cellular and tissue injury that would be expected to result in inflammation and consequent procoagulation and hypofibrinolysis.

### 3.14. Li—Lithium (Atomic Number 3, Atomic Weight 6.94)

Exposure to lithium (in the form of LiCl) commonly occurs as a medicinal agent administered as a treatment for the psychiatric illness bipolar disorder [100]. Signs and symptoms of toxicity include confusion, tremor, ataxia, lethargy, slurred speech, seizures and death [100]. Importantly, the plasma concentration of lithium in patients experiencing toxicity was noted to have a mean value of 2.16 mM, with a range of 1.5–6.7 mM [100].

LiCl appears to have been found to inhibit platelet activation in human and bovine PRP at a range of 1–50 mM by a variety of mechanisms [20,43,45,46]. However, under certain circumstances, 1–10 mM Li in PRP has been demonstrated to enhance platelet aggregation [42,44]. Lithium at 50 mM inhibited ADP-mediated human platelet activation in heparin-anticoagulated PRP and was proposed to likely compete with Ca at the cell surface or enzymatic level [20]. Further, lithium at 12.5 mM inhibited human platelet aggregation in response to thrombin, collogen, epinephrine and ristocetin, again likely secondary to competition with Ca [43]. Of interest, in the presence of ristocetin, Ca could not outcompete the effects of lithium, suggestive of another mechanism being responsible for lithium-mediated platelet inhibition [43]. Further, lithium at 1 to 5 mM inhibits vasopressin-mediated human platelet aggregation, but only in the presence of indomethacin, indicative of platelet activation via an arachidonate pathway dependent mechanism that is activated by vasopressin, independent of Ca competition [45]. Put another way, lithium inhibition of Ca-dependent platelet aggregation following vasopressin activation is of no consequence if the arachidonic acid pathway is intact, acting as the dominant pathway responsible for vasopressin-mediated platelet activation [45]. Lastly, 5 and 20 mM lithium inhibited thrombin-mediated bovine platelet aggregation by diminishing Ca mobilization within the platelet [46]. In contrast to these findings [20,43,45,46], human platelets incubated with 10 mM lithium for 90 min prior to activation demonstrated increased aggregation following arachidonic acid activation [44]. Rabbit platelets in the same study did not display the same lithium-mediated activation [44]. The proposed mechanism for enhanced aggregation was an increase in thromboxane and a decrease in adenylate cyclase activity induced by lithium [44]. Another investigation demonstrated that 1 mM lithium enhanced human platelet aggregation after a 5 min incubation with lithium followed by activation with suboptimal concentrations of ADP [42]. A reason for these differing results may be that there was no preincubation with lithium in some of the studies, with lithium being added just before platelet aggregation assessments were commenced [20,43,45,45]. It may be possible that while brief or prolonged, a 5 to 90 min incubation, respectively, may reveal a Janus-like inhibition/activation of platelet aggregation by lithium [42,44]. The reality is that in the case of therapeutic lithium plasma concentrations (0.4–0.8 mM) [100], and in the case of toxicity (2.16 mM) [100], the patient’s platelets have been constantly incubated with lithium at smaller concentrations than most of these investigations utilized [20,43,44,45,46]. In summary, with regard to platelet aggregation, it is difficult to discern the direct effects of lithium on platelet aggregation that is clinically relevant.

In the matter of coagulation, one unusual work demonstrated that the antibiotic ristocetin, a platelet activator, caused fibrinogen precipitation in purified fibrinogen solution or human plasma [47]. Lithium, at 75–300+ mM, slowed down or completely inhibited ristocetin induced fibrinogen precipitation; further, lithium addition could reverse fibrinogen precipitation after ristocetin exposure [47]. While these phenomena are difficult to relate to physiological function of fibrinogen and its interactions with platelets or thrombin, they do indicate that lithium likely binds to fibrinogen in a yet to be identified location and manner.

In conclusion, lithium may act as a platelet aggregation inhibitor or activator depending on the aforementioned conditions, and Li appears to bind to fibrinogen.

### 3.15. Mg—Magnesium (Atomic Number 12, Atomic Weight 24.30)

The normal serum magnesium concentration is 0.75–1.25 mM and it is an essential element [101], with neuromuscular and cardiac toxicity noted at 2.5–7.5 mM [101]. Toxicity usually occurs with excessive consumption, medical administration or renal failure [101].

With regard to platelet activation, exposure of washed and isolated human platelets to 1–10 mM magnesium in vitro resulted in increased adhesion to collagen and fibrinogen as well as increased P-selectin expression, but >10 mM resulted in decreased adhesion and P-selectin expression [22]. In the case of human platelets in PRP, the addition of 0.5–4.0 mM MgSO_4_ resulted in concentration-dependent decreases in aggregation in response to collagen or ADP-mediated activation [48].

As for coagulation, rabbit plasma that was decalcified demonstrated increased clotting time in response to Russel’s viper venom activation by 4–10 mM MgCl_2_ [6]. In contrast, 1 mM MgCl_2_ did not cause bovine fibrinogen-derived fibrin monomers to polymerize in a decalcified, in vitro system [7]. Humans administered MgSO_4_ in vivo demonstrated no change in intravascular coagulation, with magnesium increasing from 0.85 to 1.33 mM [49]. Further, exposure of human clotting factors FVIIa, FVIIa and FXa to up to 10 mM MgCl_2_ had no effect on the factor activity in vitro [49]. Lastly, in contrast to rabbit plasma, decalcified human plasma demonstrated a decreased prothrombin time in the presence of 0.6–2.0 mM MgCl_2_, indicative of enhancement of the TF-FVIIa-mediated coagulation [50]. However, as MgCl_2_ increased beyond 2.0 mM, prothrombin time increased [50].

In the matter of fibrinolysis, up to 6 mM MgSO_4_ had no effect on tPA-mediated thrombolysis of human blood clots in vitro [51].

In summary, physiological concentrations of magnesium are likely needed for normal platelet function and coagulation, but excessive Mg concentrations seem to primarily decrease platelet aggregation.

### 3.16. Mn—Manganese (Atomic Number 25, Atomic Weight 54.93)

Manganese is an essential element, necessary to the activity of several enzymes, including superoxide dismutase [102]. However, environmental or industrial exposure either at large doses acutely or smaller doses chronically can result in neurotoxicity manifesting as sleep disorders, muscular pain, gait changes, cephalalgia, hallucinations, and in its most severe form, a manganese-induced, Parkinson-like disease known as manganism [102].

In the matter of platelet activation, human platelet aggregation in PRP was vigorously achieved by 5 mM concentrations of manganese, with SEM demonstrating platelet ultrastructure consistent with activation [14]. Similarly, SEM of human platelets exposed to 7.3 µM manganese also demonstrated ultrastructural characteristics of activation [40]. Further, exposure of washed and isolated human platelets to 10–100 µM manganese in vitro resulted in increased adhesion to collagen and fibrinogen [22]. Lastly, a potential mechanism at play in these investigations [14,22,40], manganese is a known αIIbβ3 outside-in signaling molecule that would be expected to enhance platelet aggregation [103].

With regard to coagulation, decalcified human plasma demonstrated a decreased prothrombin time in the presence of 0.06–0.8 mM MnCl_2_, indicative of enhancement of the TF-FVIIa-mediated coagulation [50]. However, as MnCl_2_ increased beyond 1.0 mM, prothrombin time increased [50].

Lastly, as for effects of manganese on fibrinolysis, in vitro results with cultured vascular smooth muscle cells and fibroblasts demonstrated that a twenty-four hour incubation with 1 µM manganese resulted in decreased tPA and PAI-1 release from both cell types [16].

In conclusion, manganese enhances platelet activation, promotes the FVII-dependent coagulation pathway, and may decrease both profibrinolytic and antifibrinolytic enzymes from key cell types.

### 3.17. Ni—Nickel (Atomic Number 28, Atomic Weight 58.69)

Nickel exposure primarily occurs by ingestion of food or exposure to nickel-plated jewelry, with inhalation of nickel observed in industrial settings [104]. Toxicity can be manifested as contact dermatitis or as the development of lung cancer [104].

The effects of nickel on human platelet activation vary a great deal by the microenvironment within which experimentation is performed. In PRP, 5mM nickel did not affect platelet aggregation or ultrastructure as assessed by SEM with Ca present [14]. In contrast, isolated, washed platelets in Ca-free buffer demonstrated increased adherence to plastic surfaces or plastic surfaces covered in albumin after exposure to nickel; however, the work did not note the concentration of nickel or extent of adherence [22]. SEM of whole blood demonstrated increased pseudopodia formation by platelets exposed to 0.34 µM nickel [39]. Further, 1–5 mM NiCl_2_ increased platelet aggregation in in PRP with Ca absent in a albumin and nickel containing buffer in the presence of fibrinogen, but not in the absence of fibrinogen [52]. Lastly, nickel was found to enhance glycoprotein Ib-V-IX complex interactions and glycoprotein IIb/IIIa receptor formation by the platelet [52].

With regard to the effects nickel has on fibrinolysis, there are no direct works documenting modification of individual enzymes in isolation, but instead there is evidence as to what nickel does to cells in culture [16,53,54,55]. In vitro results with cultured vascular smooth muscle cells and fibroblasts demonstrated that a twenty-four hour incubation with 1 µM NiCl_2_ resulted in decreased tPA and PAI-1 release from both cell types, with tPA release inhibition much greater than PAI-1 release inhibition [16]. Human airway epithelial cells exposed to 0.58–2.34 µg/cm^2^ Ni_3_S_2_ demonstrated a decrease in urinary plasminogen activator (uPA) release without affecting cellular protein enzyme content, activity, or uPA messenger ribonucleic acid (mRNA); however, PAI-1 transcription was increased, resulting in prolonged increases in both mRNA and enzyme concentrations [53]. Lastly, the Ni_3_S_2_ induced increase in PAI-1 in these human epithelial airway cells was found to be hypoxia-inducible factor 1α-dependent [54] and did not require the presence of reactive oxygen species [55] (Ni_3_S_2_ was 9.7 µM in both studies).

In summary, nickel will activate platelets under specific conditions, and Ni appears to favor a antifibrinolytic environment in cell culture.

### 3.18. Pb—Lead (Atomic Number 82, Atomic Weight 207.20)

Lead toxicity occurs secondary to environmental exposure to contaminated air and water; exposure to lead-containing containers, paint, and piping; and, via consumption of lead-contaminated dairy products and food [105,106]. Lead damages multiple organ systems, such as the kidney and nervous system [105,106]. Blood lead concentrations between 1.9–2.4 µM are associated with peripheral neuropathy, with brain dysfunction observed between 2.4–3.4 µM, and encephalopathy occurring with concentrations >3.9 µM [106].

Lead exposure decreased rat and human platelet aggregation in PRP in vitro [56] and induced thrombocytopenia in the rat in vivo [56]. With regard to rat platelets, ADP-mediated platelet aggregation was decreased from 0 to 68% after a 5-min exposure to Pb(C_2_H_3_O_2_)_2_ (lead acetate) with a range of concentrations from 50 µM to 2.5 mM; inhibition after collagen-mediated activation ranging from 0 to 55% with the same range of lead exposure; and, thrombin-mediated activation that ranged from 14 to 100% with the same range of lead exposure [56]. In sharp contrast, exposure of human platelets to the largest lead concentration, 2.5 mM, resulted in minimal inhibition of aggregation ranging from 16–22% [56]. As for thrombin-mediated human platelet aggregation, lead inhibited between 0–66% of activation across the 50 µM to 2.5 mM range of exposure [56]. Of interest, an epidemiological investigation of tobacco smokers (n = 2,176) that had undergone platelet aggregation studies with PRP was found to have inhibition of ADP but not thrombin-mediated platelet activation via regression analyses [57]. Importantly, the average blood lead concentration of these individuals was 0.74 µM, with a range of 0.19 to 2.7 µM [57]. When the two investigations [56,57] are compared, similar results concerning inhibition of ADP-mediated platelet activation and less inhibition of thrombin-mediated activation occurred over a 1000-fold range of lead concentrations, with only the epidemiological study [57] reporting clinically relevant blood lead concentrations that could be considered toxic [106]. In summary, lead inhibits human platelet aggregation after acute exposure to very large concentrations and chronically at concentrations associated with toxicity.

As for effects on plasmatic coagulation, a 5 min incubation of human plasma with a lethal concentration of Pb(C_2_H_3_O_2_)_2_ (25 µM) did not significantly change coagulation kinetics compared to plasma without lead addition as determined by thrombelastography [38].

With regard to fibrinolysis, results with cultured vascular smooth muscle cells and fibroblasts demonstrated that a twenty-four hour incubation with 1 µM Pb(NO₃)₂ resulted in decreased tPA and PAI-1 release from both cell types, with tPA release inhibition much greater than PAI-1 release inhibition [16].

Taken as a whole, the aforementioned investigations reveal that, at clinically encountered concentrations, lead inhibits platelet aggregation, does not affect plasmatic coagulation, and may promote an antifibrinolytic environment in vitro.

### 3.19. Pt—Platinum (Atomic Number 78, Atomic Weight 195.08)

Platinum toxicity occurs primarily as a result of being administered platinum-based chemotherapeutic agents, such as carboplatin (C_6_H_12_N_2_O_4_Pt) and cisplatin (Pt(NH_3_)_2_Cl_2_, see Figure 1A) [107]. These compounds cause dose-dependent nephrotoxicity, ototoxicity, gastrointestinal toxicity and peripheral neuropathy [107].

Human plasmatic coagulation was found to not be affected by a clinically concentration (50 µM) of carboplatin diluted in either water or phosphate-buffered solution (PBS) as determined by thrombelastography [58]. However, while a 50 µM of cisplatin diluted in water did not affect plasma coagulation kinetics in the same system, cisplatin diluted in PBS delayed the time to maximum velocity of clot formation by 29% [58]. Thus, it appears that a phosphate salt of cisplatin inhibits contact protein-mediated human plasmatic coagulation [58].

### 3.20. Rb—Rubidium (Atomic Number 37, Atomic Weight 85.47)

Rubidium salts were used as a treatment in the past for depression, with a concern that renal toxicity would occur during chronic therapy [108].

In the matter of platelet activation, 50 mM RbCl significantly enhanced ADP-mediated platelet aggregation in human PRP that was anticoagulated with heparin [20]. ADP-mediated human platelet aggregation following exposure to 1 mM rubidium was also noted [41]. Thus, rubidium enhances ADP-mediated human platelet aggregation.

### 3.21. Ru—Ruthenium (Atomic Number 44, Atomic Weight 101.07)

Ruthenium has been used in the form of ruthenium red (H_42_C_l6_N_14_O_2_Ru_3_) to induce neurotoxicity in cell culture [109]; further, ruthenium-containing chemotherapeutic agents, such as NAMI-A (C_8_H_15_C_l4_N_4_ORuS), have undergone clinical trials in recent years, with gastrointestinal and hematological toxicity noted [110].

Utilizing thrombelastography, it was determined in human plasma that NAMI-A dissolved in water prior to being placed in plasma (50 µM final concentration) increased the velocity of clot formation compared to plasma not exposed to NAMI-A—a result that did not occur when NAMI-A was dissolved in PBS [58]. In contrast, RuCl_3_ dissolved in PBS placed in human plasma (50 µM final concentration) demonstrated strong procoagulant effects, with the time to reach maximum velocity of clot formation decreased by 50% and the velocity of clot formation increased more than two-fold compared to plasma without RuCl_3_ addition [58]. This procoagulant enhancement of coagulation was not observed if RuCl_3_ was dissolved in water [58]. By utilizing an array of different coagulation activators and coagulation factor-depleted plasmas, it was determined that RuCl_3_ dissolved in PBS was enhancing prothrombin activation [58]. In conclusion, two ruthenium-containing compounds under specific circumstances appear to enhance human plasmatic coagulation, with RuCl_3_ likely interacting with a PO_4_ ion to cause increased prothrombin activation.

### 3.22. Sc—Scandium (Atomic Number 21, Atomic Weight 44.96)

Scandium exposure occurs from environmental and industrial sources [111,112], with acute scandium exposure in rats compromising renal function [110], and with patients with compromised renal function noted to have increased plasma scandium concentrations [112]. Normal human subjects were noted to have 0.64 nM scandium blood concentrations, whereas patients with renal disease had 1.21 nM scandium present in their circulation [112].

Using a rat model, ScCl_3_ was utilized to assess changes in coagulation in vitro and in vivo [59]. Whole blood anticoagulated with sodium oxalate had ScCl_3_ added for a final concentration of 22.2 mM, resulting in a doubling of prothrombin time, a more than 5-fold increased activated partial thromboplastin time, and doubling thrombin time compared to blood not exposed to ScCl_3_ [59]. Rats weighing 200 gm (with presumed circulating blood volumes of 70 mL/kg or 14 mL) were administered 6 mg of ScCl_3_ intravenously, resulting in an instantaneous blood concentration of 0.43 mg/mL or 9.6 mM [59]. Compared to rats not administered ScCl_3_, rats injected with ScCl_3_ had a 50% prolongation of prothrombin time, doubling of activated partial thromboplastin time, and a more than 5-fold increase in thrombin time [59]. After 24 h, rats injected with ScCl_3_ had no significant difference in these coagulation parameters compared to ScCl_3_-naïve rats [59]. In conclusion, blood concentrations of scandium many orders of magnitude than that observed clinically in humans appear to reversibly inhibit whole blood coagulation activated by either contact protein- or tissue factor-mediated pathways in rats.

### 3.23. Sr—Strontium (Atomic Number 38, Atomic Weight 87.62)

Humanity is exposed to strontium from the crust of the earth, it being ubiquitous in water, and generally ingested through the gastrointestinal tract and stored in bone [113,114]. A human case of strontium toxicity has not been observed, and in most normal people it accounts for 0.00044% of body mass (Ca accounts for 1.4% of body mass) [113,114].

The effect of strontium on platelets has been tested in a variety of mammalian species at several concentrations [5,14,60,61]. Strontium activates and inhibits rat PRP platelet aggregation induced with thrombin in a Ca-free buffer in a biphasic manner, with an increase in aggregation observed between 0.33 to 2 mM followed by a loss of aggregation from 2 mM to 20 mM SrCl_2_ [5]. In the case of human PRP in Ca-free buffer, 5 mM strontium did not cause platelet aggregation [14]. Similar to rat platelets, rabbit PRP in Ca-free buffer demonstrated increased platelet aggregation in the presence of 0.5 to 2 mM SrCl_2_ [61]. Lastly, ovine PRP that was anticoagulated with heparin demonstrated no increase in platelet factor 4 release in response to activation by collagen, ADP or thrombin after exposure to 5 mM or 100 mM strontium compared to samples not exposed to strontium [60].

As for coagulation, the time to coagulation of rabbit plasma was rapidly decreased from 380 s to 100 s by 1 mM SrCl_2_, decreasing to 80 s with 2 mM SrCl_2_ and remaining plateaued at that value up to 3.5 mM concentrations after oxalate anticoagulation [6]. A similar pattern was observed with experiments involving rabbit plasma anticoagulated with sodium citrate that utilized up to 8 mM SrCl_2_ [6]. Lastly, purified bovine fibrinogen in Ca-free buffer did not demonstrate significant fibrin monomer polymerization when exposed to 1 mM Sr [7].

In conclusion, strontium appears to activate platelet aggregation and plasmatic coagulation at concentrations of 1 to 2 mM but decreases platelet aggregation at greater concentrations—which are strontium concentrations not seen in vivo.

### 3.24. V—Vanadium (Atomic Number 23, Atomic Weight 50.94)

Exposure to vanadium primarily occurs via inhalation, with vanadium being released following combustion of coal and other fossil fuels and industrial sources, with very poor adsorption from the gastrointestinal tract [115,116,117]. Neurotoxicity is the primary adverse outcome after prolonged exposure to vanadium from the environment or from nutritional/antiglycemic supplements [115,116,117]. A typical vanadium compound encountered via inhalation includes vanadium pentoxide (V_2_O_5_), which is rapidly converted to other vanadium containing compounds that react with biomolecules [117].

With regard to platelet activation, exposure of mice in vivo and human platelets as PRP in vitro to V_2_O_5_ was performed [62]. The mice were exposed to a 20 mM V_2_O_5_ aerosol in a chamber for one hour a day, twice a week for eight weeks [62]. Human PRP was acutely exposed to 5, 50, 500 and 5000 µM V_2_O_5_ [62]. The rats demonstrated a 51% inhibition of ADP-mediated platelet aggregation after 4 weeks of V_2_O_5_ exposure, but inhibition decreased to less than 8% by eight weeks of exposure [62]. Human platelets demonstrated a dose-dependent inhibition of platelet aggregation of approximately 10% to 66% following exposure to 5 and 5000 µM V_2_O_5_, respectively [62].

In summary, it appears that vanadium is an inhibitor of platelet aggregation following acute but perhaps not chronic exposure.

### 3.25. Zn—Zinc (Atomic Number 30, Atomic Weight 65.38)

Zinc is an essential element. Exposure occurs via the lung, skin, and respiratory tract, with sources including industrial gaseous release, inhalation of smoke from fireworks, and vitamin supplementation [118,119]. Thus, pulmonary injury, gastrointestinal distress, and neurological symptoms such as lethargy and focal neurological deficits have been observed secondary to zinc toxicity [118,119].

With regard to the effects of zinc on platelet activation, exposure of washed and isolated human platelets to 100 µM–1.4 mM zinc (physiological range) in vitro resulted in increased adhesion to collagen and fibrinogen, but exposure to >10 mM decreased platelet adhesion [22]. As recently reviewed in detail, zinc is critical to platelet activation and is dependent on protein kinase C activity and associated with modulation of tyrosine phosphatase activity [63]. Zinc acts as a second messenger in other pathways, causing changes in platelet morphology and assembly of glycoprotein IIb/IIIa on the platelet surface [63]. For greater detail, the reader is invited to examine the aforementioned review [63].

As for zinc-mediated modulation of plasmatic coagulation, it is complex and involves many points in the various pathways [7,64,65,66,67]. In an early work, zinc (6.67 µM, optimal range 10–16 µM in vitro) can trigger factor XII activation without a surface present, which in turn will cause activation of the remainder of the contact protein pathway via activated factor XII [64]. However, zinc decreases the amidolytic activity of activated factor VII and activated factor X [66]. Also of interest, compared to several other divalent metals, 1 mM ZnCl_2_ caused bovine fibrinogen-derived fibrin monomers to polymerize to the greatest extent in a decalcified, in vitro system [7]. Zinc (4–20 µM) diminished thrombin-mediated clotting time in human fibrinogen solutions containing factor XIII, with gel permeability and microscopy studies demonstrating that the resulting clots were more porous and had thicker fibrin polymers with less crosslinking in the absence or presence of Ca [65]. Of interest, when Ca was absent and zinc was present, there was no activated factor XIII-mediated crosslinking observed [65]. In a similar study several years later, 0–5 µM zinc was found to decrease the time to clotting in a concentration-dependent fashion; further, thickened fibrin polymers with increasing zinc concentrations were observed, as well as loss of clot strength determined with a viscoelastic, rheological method [67]. Critically, the increase in clot stiffness caused by factor XIII-mediated crosslinking of fibrin polymers was diminished by zinc, independent of factor XIII activity [67]. In summary, zinc both enhances and diminishes several aspects of coagulation, and the reader is directed to an excellent review of these processes for more detail [66].

In the matter of zinc-mediated effects on fibrinolysis, the metal displays both profibrinolytic and antifibrinolytic modulation [36,66,68,69,70]. For example, 10 µM zinc inhibited thrombolysis of murine whole blood clots by tPA (origin of tPA not provided) [36]. In contrast, 50–1000 µM zinc enhanced binding of human single chain uPA to fibrin polymers, which would be expected to enhance fibrinolysis [68]. However, the antifibrinolytic enzyme, thrombin-activatable fibrinolysis inhibitor (TAFI), that removes lysine residues from partially lysed fibrin polymer and thus attenuates tPA-mediated plasminogen activation is a zinc dependent enzyme—and without zinc bound to it, TAFI cannot be activated [69]. Lastly, in a system of human enzymes, 0.5–5 µM zinc enhances the affinity of tPA binding to fibrin polymers; however, zinc also decreases tPA-mediated activation of plasminogen 2-fold, resulting in a 2.5-fold prolongation of clot lysis time of fibrin clots and a 1.5-fold prolongation of clot lysis time in plasma thrombi [70]. In conclusion, zinc is necessary for some antifibrinolytic enzymes to function, and in other instances Zn will diminish fibrinolysis by decreasing profibrinolytic enzymatic activity.

Taken as a whole, zinc appears to enhance activation of platelets, promote faster developing plasma clots that have weaker viscoelastic properties, and likely result in a milieu wherein fibrinolysis is attenuated.

To assist the readership in associating the effects of all the aforementioned Figure 2, Figure 3 and Figure 4 are now presented, which involve the processes of platelet activation, coagulation, and fibrinolysis, we enclose three figures that summarize these relationships.

**Figure 1 ijms-24-03302-f001:**
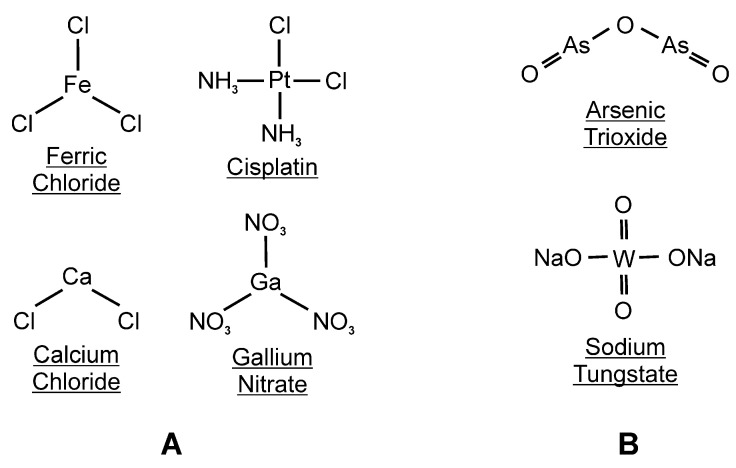
Structures of various metal-containing compounds. Panel (**A**)—the metal cation contained in these four compounds is likely to directly bind to protein targets. Panel (**B**)—the metalloid and metal depicted in these two molecules are not likely to directly interact with proteins. As = arsenic, Ca = calcium, Cl = chloride, Fe = iron, Ga = gallium, Na = sodium, NH_3_ = ammonia, NO_3_ = nitrate, O = oxygen, Pt = platinum and W = tungsten.

**Figure 2 ijms-24-03302-f002:**
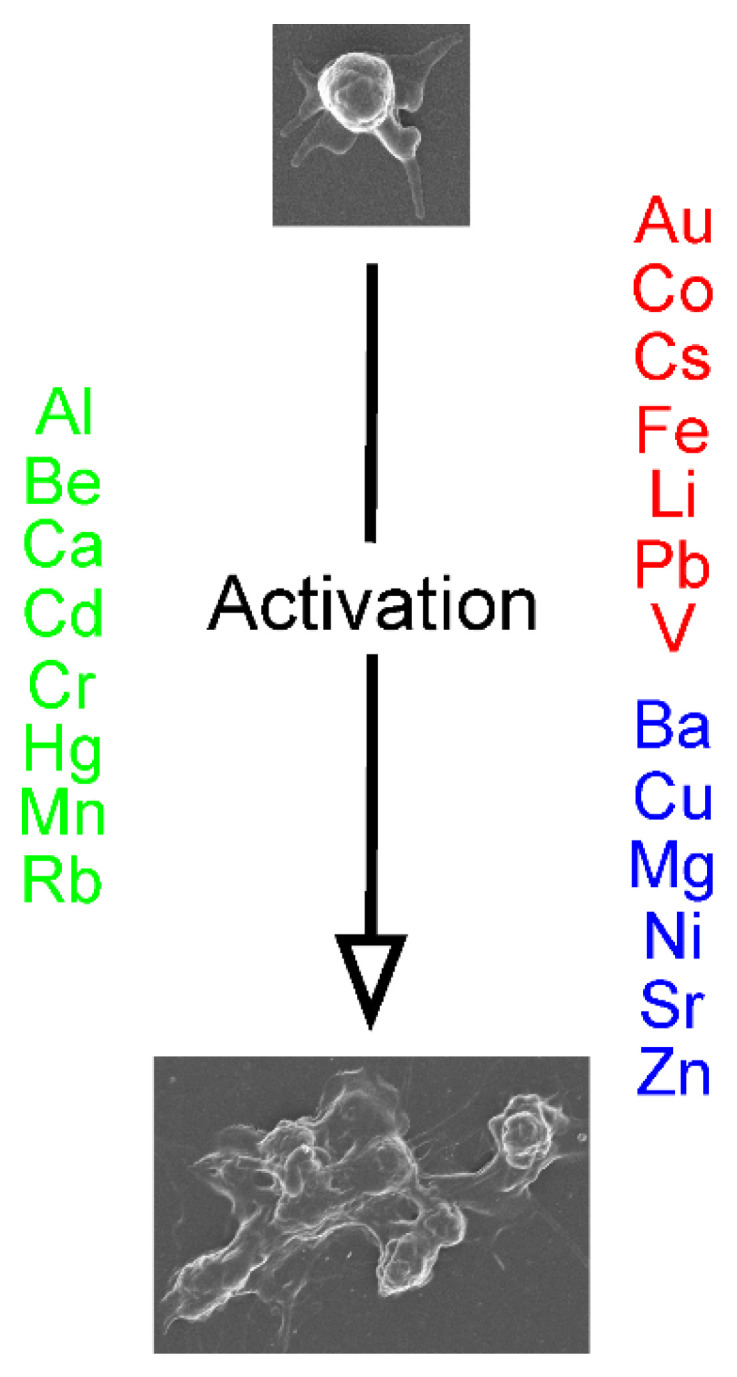
Effects of metals on platelet activation. Metals depicted in green enhance activation, those displayed in red decrease activation, and elements that are blue have conflicting effects on platelet activation that may be concentration-dependent or dependent on the species source of the platelets.

**Figure 3 ijms-24-03302-f003:**
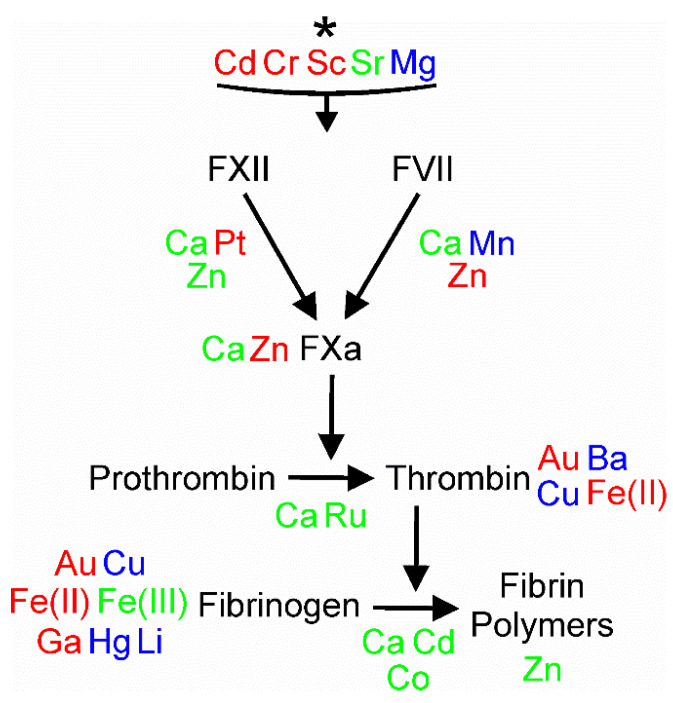
Effects of metals on coagulation. Metals depicted in green enhance coagulation, those displayed in red decrease clotting, and elements that are blue have conflicting effects on coagulation that may be concentration-dependent or dependent on the species source of the plasma or blood. Metals seen at the top of the figure with the * are those without a specific site in the coagulation pathways identified. All metals have been defined in the text, but the depiction of iron was changed to improve the visual presentation. Fe^+2^ = Fe(II) and Fe^+3^ = Fe(III). Please refer to the text for greater details about the specific coagulation enzymes and pathways depicted.

**Figure 4 ijms-24-03302-f004:**
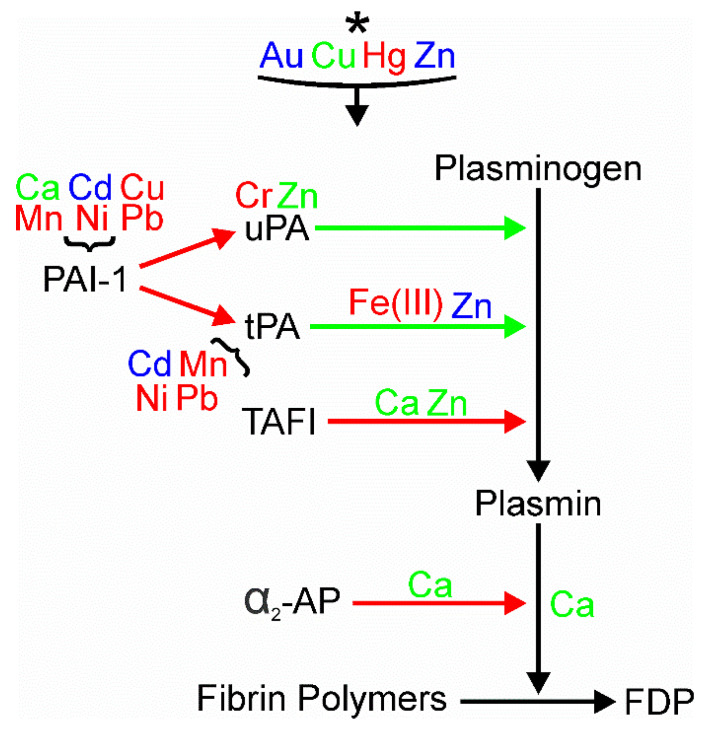
Effects of metals on fibrinolysis. Metals depicted in green enhance fibrinolysis, those displayed in red decrease clot dissolution, and elements that are blue have conflicting effects on fibrinolysis that may be concentration-dependent or dependent on the species source of the plasma or blood. Metals seen at the top of the figure with the * are those without a specific site in the fibrinolytic pathways identified. Green arrows indicate that the enzyme increases the indicated reaction and red arrows indicate that the enzyme inhibits the pathway indicated. All metals have been defined in the text, but the depiction of iron was changed to improve the visual presentation. Fe^+2^ = Fe(II) and Fe^+3^ = Fe(III). Please refer to the text for greater details about the specific fibrinolytic enzymes and pathways depicted.

## 4. Conclusions

As can be appreciated from the data presented, metals interact in potentially very critical ways with nearly all aspects of platelet activation, plasmatic coagulation, and fibrinolysis. It is our intention that this work serve not as an ending point, but rather as a fair evaluation of what mechanisms concerning metal interactions with the hemostatic system have been elucidated, and as a beacon to guide future investigation. Further study will be needed to better characterize the pathological and clinically-relevant effects of metals on hemostasis.

## Data Availability

Not applicable.

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
