# Peer review of "The Gilded Clot: Review of Metal-Modulated Platelet Activation, Coagulation, and Fibrinolysis"

_ijms, 2023, doi:10.3390/ijms24043302_

Round 1

Reviewer 1 Report

I have the following suggestions/comments:

1. The authors appear to have turned off spell check, so there are several spelling errors that need to be fixed:

(a) page 5: condtions (conditions)

(b) page 6: beteen (between)

(c) page 7: suppliments (supplements)

(d) page 7: complimentary (complementary)

(e) page 7: ultrasturctural (ultrastructural)

(f) page 7: reversable (reversible)

(g) page 8: procoagulate (procoagulant)

(h) page 12: inhibted (inhibited); concenetrations (concentrations)

(i) page 13: disolved (dissolved)

2. Some sections of the manuscript are unclear or potentially represent incomplete sentences:

(a) page 4: "Ca is required to microtubular formation" should read "Ca is required for microtubular formation"

(b) page 5: "However, when PRP obtained from an afibrinogenemic patient was exposed to Co for 90 minutes had addition of fibrinogen, aggregation was restored" would be better as "However, when PRP obtained from an afibrinogenemic patient was exposed to Co for 90 minutes and had fibrinogen added, aggregation was restored" - also, would this not have occurred without addition of Co? So, what was special about the addition of Co?

(c) page 5, staring line 241: "Cr can be found in drinking water and in industrial industrial settings involving is widely used in numerous industrial processes" suggest "Cr can be found in drinking water and in industrial settings involving numerous processes"

(d) Page 6: line 295: "were noted to polymerized" should read "were noted to polymerize"

(e) page 8: "...found to exert local hemostasis..." ("...found to exert local hemostasis effects...")

(f) page 8: "...period the animals had a marked increase in coagulation, with time to onset of coagulation, velocity of clot growth and clot strength
significantly and markedly changed" ("...marked increase in coagulation..." is missing something)

(g) page 10: "Mn-inducedParkinson-like" missing space

(h) page 11: "...tPA release inhibition much greater than PAI-1 release..." ("...tPA release inhibition much greater than PAI-1 release inhibition...")

(i) page 12: "...with tPA release inhibition much greater than PAI-1 release..." ("...with tPA release inhibition much greater than PAI-1 release inhibition...")

(j) page 12, line 589: "...in vitro" ("...in vitro.")

(k) page 12: "...by a clinically concentration..." ("...by a clinically relevent concentration...")

3. The authors abbreviate the metals to use the abbreviated term; this makes sense when the authors use formula, but makes the reading itself more difficult. In general, it is not easier to read an abbreviation of a single word than the word itself; I would support using the full term in sentences, and keeping the abbreviations only for the formula. For example (page 2): "...with As or W as displayed in figure 1, panel B. Thus, while both As and W... " "Thus, as with As and W, Bi is most likely..." requires mental effort to convert the abbreviations back to the metal in question.

4. Page 7: "human plasminogen activator type 1 (PAI-1)" PAI-1 was already abbreviated on page 4, and used several times from page 4.

5. page 13: "The effect of Sr on platelets has been tested in a variety of mamalian species at several concentrations [5,14,60,61]. Sr activates and inhibits rat PRP platelet aggregation in Ca free buffer in a biphasic manner, with an increase in aggregation observed between 0.33 to 2 mM followed by a loss of aggregation from 2 mM to 20 mM SrCl2 [5]. In the case of human PRP in Ca free buffer, 5 mM Sr did not cause platelet aggregation." Is this aggregation caused by SR, or by an agonist(s) and which Sr then inhibits or augments?

6. Page 14: "...can trigger factor XII activation without a surface present, which in turn will cause activation of the remainder of the contact protein
pathway via activated factor VII" How so? FXII activation and FVII activation represent different pathways.

7. Page 15: "...resulting clots were more porous and had thicker fibrin polymers with less crosslinking in the absence or presence of Ca [65]. Of interest, when only Zn was present, there was no activated factor XIII mediated crosslinking observed" unclear; "less crosslinking in the absence or presence of Ca" vs "no crosslinking when only Zn was present" (suggesting in the absence of Ca).

Author Response

We greatly appreciate the time the reviewer invested in our manuscript. We were unable to restore the spelling function of the template provided by the journal and tried our best to identify misspellings. Even with the new template with the revised manuscript sent by the publisher, the spell checker missed all these errors kindly identified by the reviewer. We address each error as indicated.

I have the following suggestions/comments:

  1. The authors appear to have turned off spell check, so there are several spelling errors that need to be fixed:

(a) page 5: condtions (conditions) – corrected.

(b) page 6: beteen (between) – corrected.

(c) page 7: suppliments (supplements) – corrected.

(d) page 7: complimentary (complementary) – corrected.

(e) page 7: ultrasturctural (ultrastructural) – corrected.

(f) page 7: reversable (reversible) – corrected.

(g) page 8: procoagulate (procoagulant) – corrected.

(h) page 12: inhibted (inhibited); concenetrations (concentrations) – corrected.

(i) page 13: disolved (dissolved) – corrected.

  1. Some sections of the manuscript are unclear or potentially represent incomplete sentences:

(a) page 4: "Ca is required to microtubular formation" should read "Ca is required for microtubular formation" – The modification was made.

(b) page 5: "However, when PRP obtained from an afibrinogenemic patient was exposed to Co for 90 minutes had addition of fibrinogen, aggregation was restored" would be better as "However, when PRP obtained from an afibrinogenemic patient was exposed to Co for 90 minutes and had fibrinogen added, aggregation was restored" – This suggested modification was made.

…also, would this not have occurred without addition of Co? So, what was special about the addition of Co?

Thank you for this insightful comment. In the experiment with PRP from normal subjects, the fibrinogen and platelets were exposed to Co for 90 minutes. In the afibrinogenemic patient, only their platelets were exposed for 90 minutes while the exogenous fibrinogen that was Co-naïve was added just before the reaction occurred. So, Co did not have an opportunity to bind for 90 minutes with the fibrinogen, and the normal response with the afibrinogenemic PRP demonstrates that it is a Co mediated modification of fibrinogen, not platelets, responsible for the phenomena observed. We have changed the text to clarify this issue.

(c) page 5, staring line 241: "Cr can be found in drinking water and in industrial industrial settings involving is widely used in numerous industrial processes" suggest "Cr can be found in drinking water and in industrial settings involving numerous processes"

The suggested modification was made.

(d) Page 6: line 295: "were noted to polymerized" should read "were noted to polymerize"

The text was changed.

(e) page 8: "...found to exert local hemostasis..." ("...found to exert local hemostasis effects...")

We changed this to hemostatic effects.

(f) page 8: "...period the animals had a marked increase in coagulation, with time to onset of coagulation, velocity of clot growth and clot strength

significantly and markedly changed" ("...marked increase in coagulation..." is missing something)

We have broken this sentence into two, which should increase clarity.

(g) page 10: "Mn-inducedParkinson-like" missing space – corrected.

(h) page 11: "...tPA release inhibition much greater than PAI-1 release..." ("...tPA release inhibition much greater than PAI-1 release inhibition...") – corrected.

(i) page 12: "...with tPA release inhibition much greater than PAI-1 release..." ("...with tPA release inhibition much greater than PAI-1 release inhibition...") – corrected.

(j) page 12, line 589: "...in vitro" ("...in vitro.")

(k) page 12: "...by a clinically concentration..." ("...by a clinically relevent concentration...")

  1. The authors abbreviate the metals to use the abbreviated term; this makes sense when the authors use formula, but makes the reading itself more difficult. In general, it is not easier to read an abbreviation of a single word than the word itself; I would support using the full term in sentences, and keeping the abbreviations only for the formula. For example (page 2): "...with As or W as displayed in figure 1, panel B. Thus, while both As and W... " "Thus, as with As and W, Bi is most likely..." requires mental effort to convert the abbreviations back to the metal in question. – We have complied with this request.

  1. Page 7: "human plasminogen activator type 1 (PAI-1)" PAI-1 was already abbreviated on page 4, and used several times from page 4. – corrected.

  1. page 13: "The effect of Sr on platelets has been tested in a variety of mamalian species at several concentrations [5,14,60,61]. Sr activates and inhibits rat PRP platelet aggregation in Ca free buffer in a biphasic manner, with an increase in aggregation observed between 0.33 to 2 mM followed by a loss of aggregation from 2 mM to 20 mM SrCl2 [5]. In the case of human PRP in Ca free buffer, 5 mM Sr did not cause platelet aggregation." Is this aggregation caused by SR, or by an agonist(s) and which Sr then inhibits or augments?

Aggregation was induced with thrombin and modulated by Sr. We have added this to the text.

  1. Page 14: "...can trigger factor XII activation without a surface present, which in turn will cause activation of the remainder of the contact protein

pathway via activated factor VII" How so? FXII activation and FVII activation represent different pathways.

The reviewer is correct. We meant activated factor XII, not VII. We have corrected the error.

  1. Page 15: "...resulting clots were more porous and had thicker fibrin polymers with less crosslinking in the absence or presence of Ca [65]. Of interest, when only Zn was present, there was no activated factor XIII mediated crosslinking observed" unclear; "less crosslinking in the absence or presence of Ca" vs "no crosslinking when only Zn was present" (suggesting in the absence of Ca).

We have modified the sentence to clarify this issue.

Reviewer 2 Report

This review focused on effects of metals on platelet activity, plasmatic coagulation, and fibrinolysis. The problem itself is important and information concerning different metals action on blood coagulation system is very limited. The review contain important information concerning many metals effects on blood, however critical evaluation of the presented results are often missing. In the most of cited papers, extremely high concentrations of the metals are used for evaluation of platelet activity, plasmatic coagulation, and fibrinolysis. Only in a few cases, the authors indicated possible concentrations in blood during chemotherapy or poisoning. I will suggest to authors to include more information concerning real concentrations of the metals in blood under different conditions.

For example “With regard to effects on human platelets, CsCl (50 mM) signifiantly inhibited ADP mediated aggregation of heparin anticoagulated PRP [20]”.  50 mM is extremely high concentration and it might be some unspecific effects on platelets not directly related to Cs.

The same with Ga. Toxicity from prolonged infusion of Ga(NO3)3 involves the intestine, manifesting as diarrhea [98]. However, the concentration of Ga in blood is not indicated and then “further 39-50 mM Ga(NO3)3 removed clottable fibrinogen in human plasma [37]. 39 – 50 mM. What was the reason to use such a high concentrations? In all such cases the authors should be more critical concerning presented data and if possible include more information concerning mechanisms of action of investigated metals.

Mn. Mn is a known integrin αII/βIII integrin outside-in signaling in platelets and such information should be added.

V – vanadium. It is known that Vanadate and pervanadate are commonly used general protein-tyrosine phosphatase (PTP) inhibitors. Such information also will be good to add to the review.

Minor.

P. 14, l 707. “protein kinase c activity”. C should be capital C.

Author Response

This review focused on effects of metals on platelet activity, plasmatic coagulation, and fibrinolysis. The problem itself is important and information concerning different metals action on blood coagulation system is very limited. The review contain important information concerning many metals effects on blood, however critical evaluation of the presented results are often missing. In the most of cited papers, extremely high concentrations of the metals are used for evaluation of platelet activity, plasmatic coagulation, and fibrinolysis. Only in a few cases, the authors indicated possible concentrations in blood during chemotherapy or poisoning. I will suggest to authors to include more information concerning real concentrations of the metals in blood under different conditions.

We appreciate this comment and wish the reviewer to know that this is exactly what we did when writing this manuscript. We made great effort to identify when possible the information suggested. We now include the following paragraph at the end of section 2:

“As a final note before the metals are presented, a significant effort was made to determine what, if any, blood, or plasma concentrations of the metals are present in the settings of normal living or poisoning. In the case of the in vitro studies, the concentrations utilized by investigators in decades past may seem very large and may not correspond to reasonable in vivo or cellular concentrations. Nevertheless, the authors hold that presentation of what is known is of greater benefit than post hoc judging experimentation from the past.”

For example “With regard to effects on human platelets, CsCl (50 mM) signifiantly inhibited ADP mediated aggregation of heparin anticoagulated PRP [20]”.  50 mM is extremely high concentration and it might be some unspecific effects on platelets not directly related to Cs.

We appreciate this comment and refer the reviewer to the preceding response we made and new paragraph. As for this particular study, the CsCl was at the indicated concentration, and whatever the effect observed (modulation, cytotoxicity, etc.), it is CsCl causing it. Again, our goal is to present what is known and we cannot post hoc determine more appropriate experiments.

The same with Ga. Toxicity from prolonged infusion of Ga(NO3)3 involves the intestine, manifesting as diarrhea [98]. However, the concentration of Ga in blood is not indicated and then “further 39-50 mM Ga(NO3)3 removed clottable fibrinogen in human plasma [37]. 39 – 50 mM. What was the reason to use such a high concentrations? In all such cases the authors should be more critical concerning presented data and if possible include more information concerning mechanisms of action of investigated metals.

Thank you for this comment. We could find the indicated article about toxicity but there were no blood concentrations. Again, it is entirely reasonable to wonder about the concentrations in vitro used, but in the absence of prospective work, this is all we can present. We provide whatever information we found, and included whatever degree of mechanism elucidated by the authors.

Mn. Mn is a known integrin αII/βIII integrin outside-in signaling in platelets and such information should be added.

We thank the reviewer for this insight. We have added the information and include a new reference 103.

V – vanadium. It is known that Vanadate and pervanadate are commonly used general protein-tyrosine phosphatase (PTP) inhibitors. Such information also will be good to add to the review.

We appreciate this comment, but the compounds cited by the author are like sodium tungstate wherein a direct interaction of the metal with proteins is not going to happen. We make this comment based on the molecular structure of these compounds. Thus, given our exclusion criteria, we prefer not to mention these compounds.

Minor.

  1. 14, l 707. “protein kinase c activity”. C should be capital C.

Corrected.